# Incidence of Type 1 Diabetes in Children Aged 0–14 Years in Trentino–Alto Adige Region and Determinants of Onset with Ketoacidosis

**DOI:** 10.3390/jpm14101055

**Published:** 2024-10-11

**Authors:** Stefania Fanti, Denise Lazzarotto, Petra Reinstadler, Nadia Quaglia, Evelina Maines, Maria Agostina Lamberti, Vittoria Cauvin, Riccardo Pertile, Massimo Soffiati, Roberto Franceschi

**Affiliations:** 1Department of Pediatrics, Santa Chiara Hospital of Trento, Azienda Provinciale per i Servizi Sanitari della Provincia Autonoma di Trento, 38122 Trento, Italy; stefania.fanti@apss.tn.it (S.F.); nadia.quaglia@apss.tn.it (N.Q.); vittoria.cauvin@apss.tn.it (V.C.); massimo.soffiati@apss.tn.it (M.S.); roberto.franceschi@apss.tn.it (R.F.); 2Ospedale di Bolzano, Azienda Sanitaria dell’Alto Adige, 39100 Bolzano, Italy; denise.lazzarotto@sabes.it (D.L.); petra.reinstadler@sabes.it (P.R.); mariaagostina.lamberti@sabes.it (M.A.L.); 3Department of Clinical and Evaluative Epidemiology, Azienda Provinciale per i Servizi Sanitari della Provincia Autonoma di Trento, 38122 Trento, Italy; riccardo.pertile@apss.tn.it

**Keywords:** ketoacidosis, incidence, epidemiology, pediatric, T1D

## Abstract

Aim: To assess the incidence and the temporal trend of type 1 diabetes (T1D) and diabetic ketoacidosis (DKA) during the period 2014–2023 in youths aged 0–14 years in the Trentino–Alto Adige region, Italy. Methods: A retrospective review of all incident cases of T1D diagnosed at the two Pediatric Diabetes Centers of Bolzano and Trento was matched with diabetes exemptions (No. 344). Demographic, clinical, and socioeconomic status (SES) data at first hospitalization were collected from subjects who agreed to participate (No. 272). Results: The incidence of T1D was 21.5/100,000 person/years, with a peak of 31.1 in 2021 during the COVID-19 pandemic. The mean age at the onset was 8.8 ± 3.9 years. Seventy-nine percent of the subjects were Italians, primarily residents in rural areas, and SES was equally represented. The mean incidence of DKA was 36.9%. The logistic regression analysis showed that the independent characteristics of the patients with DKA were of a younger age and displayed higher glycated hemoglobin (HbA1c) values. No relation of DKA with seasonality, ethnicity, or first-degree relative (FDR) with T1D or SES was detected. Conclusions: Our study revealed an incidence of T1D in the Trentino–Alto Adige region comparable to other areas in the North of Italy. The DKA rate negatively correlated with age; therefore, targeted prevention educational campaigns to increase awareness are needed.

## 1. Introduction

The worldwide incidence of type 1 diabetes (T1D) has increased in recent decades, and overall, approximately 1.2 million children and adolescents younger than 20 years are currently affected by the disease globally [1]; every year, around 110,000 new diagnoses in children under 15 years are estimated [2]. The incidence has increased in recent decades at a rate of approximately 3% per annum in several countries, although it seems to have reached a plateau in the last few years [2].

Significant geographical variation in the incidence of childhood T1D is observed between different countries and within countries themselves. The reasons for this variability are still unknown. However, genetic and non-genetic factors like environment and lifestyle changes may play a role in this complex process [3]. The highest rates were registered in Northern Europe (52.2 per 100,000 in Finland), while Asia appeared to have the lowest incidence (from 1.9 to 2.2 per 100,000 person-years in China and Japan) [4]. Italy seems to share the intermediate–high risk for T1D seen in other European countries, varying, in youths 0–14 years, from 20 per 100,000 in Veneto and Calabria [5,6] up to 73.9 per 100,000 in Sardinia [7]. From data available in the literature, higher rates were generally registered in northern Italy, except for the island of Sardinia, which has the second highest incidence of T1D in the world [5,6,7,8,9,10,11]. In addition, a global pattern of seasonality with T1D onset was found: cases tend to peak in the winter months and trough in the summer months due to unclear reasons, possibly linked to viral etiology [12]. The COVID-19 pandemic caused an increase in the number of new cases of T1D compared to that in the pre-COVID-19 period, and also the frequency of DKA at the onset of T1D increased in some cohorts, mainly due to delayed diagnoses secondary to isolation caused by the pandemic [13].

The early diagnosis of T1D helps to reduce life-threatening complications such as diabetic ketoacidosis (DKA); hyperglycemia, metabolic acidosis, and ketonemia characterize this condition with an increased risk of cerebral edema, cognitive deficits, or even death [14,15]. Moreover, the costs of DKA-related hospital admission are high [9]. Specifically, findings from recent studies showed that countries with a higher T1D incidence have a lower frequency of DKA through increased awareness of the disease: knowledge of the critical signs or symptoms of the disease, an older age at T1D onset, higher levels of parental education, and living in bigger cities are factors associated with decreased risk of DKA [16].

In the northern Italian region of Trentino–Alto Adige, the incidence of T1D, the frequency and severity of DKA at diabetes onset, and the temporal trend in the last decade have not yet been studied. This study aims to include this information and investigate the possible factors associated with DKA. Understanding the trend in T1D incidence and identifying controllable risk factors related to DKA could help us to develop targeted and personalized interventions to reduce the rate and severity of this complication.

## 2. Materials and Methods

This study utilized the following participant criteria:− Children and adolescents aged between 1 and 14 years;− Individuals with a diagnosis of T1D between January 2014 and December 2023;− Individuals residing in the Trentino–Alto Adige region.

In Italy, starting in 1997, a Registry of Childhood Type 1 Diabetes Mellitus in Italy (RIDI) was established to coordinate regional registries. In the Trentino–Alto Adige region, no institutional regional registers are available. Only two centers belonging to the Italian Society of Pediatric Endocrinology and Diabetology (ISPED) network [17] usually receive all the new cases of youths with pediatric diabetes, the Center of Bolzano and the one in Trento, and they register the patients using electronic health records. Therefore, for this study, local internal electronic registers of cases per year were consulted and matched with the User Fee-Exempt Registry for diabetes and data from the Hospital Discharge Registry (ICD9-CM codes 250.x1 and 250.x3).

Inclusion criteria required diagnosis of T1D confirmed by positivity of at least one of the diabetes-associated autoantibodies; all the patients with secondary or monogenic diabetes were excluded. International Society for Pediatric and Adolescents diabetes (ISPAD) guidelines were followed to define DKA presence at onset and severity, from mild (pH < 7.3, bicarbonate < 15 mmol/L) and moderate (pH < 7.2, bicarbonate < 10 mmol/L) to severe (pH < 7.1, bicarbonate < 5 mmol/L) [18].

The annual T1D incidence rate was calculated using the population aged 0–14 years for the entire region and separately for each province. Data were derived from the National Institute of Statistics https://demo.istat.it/.

Clinical and demographic characteristics were collected from the patient’s clinical notes. They included date of birth, date of diabetes diagnosis, gender, place of residence, ethnicity, pH value, glycated hemoglobin (HbA1c) and bicarbonate at diabetes onset, presence and severity of DKA, and presence of first-degree relative (FDR) with T1D. The family’s socioeconomic status (SES) was evaluated using the mother’s and father’s educational levels, occupation, gross household income, home ownership, age, and ethnicity. According to this, the SES score was classified at three levels, low, medium, and high, as previously reported [19,20]. We considered residence at T1D onset to be in an urban area in the case of more than 10,000 inhabitants.

This study was carried out in accordance with the Declaration of Helsinki and approved by the Institutional Review Board of “Azienda Provinciale per i Servizi Sanitari della Provincia Autonoma di Trento” (reference number 14854/2019). Written informed consent was obtained from each participant and parent/legal guardian, as applicable, before enrollment.

### Statistical Analysis

The Kolmogorov–Smirnov test was used to assess the normal distribution of the variables. Quantitative variables were summarized using mean standard deviation (SD), while qualitative variables were summarized as absolute and percentage frequencies. The incidence of T1D was evaluated as punctual and 95% confidence interval (95% CI). The chi-squared goodness-of-fit test was used to test whether the observed distribution of the categorical variables (gender, age group, and season of T1D diagnosis) differs from that of the expected distribution. Differences between patients with and without DKA at T1D diagnosis were performed using the Student t-test or the Kruskal–Wallis test (in case of not normally distributed variables) for quantitative variables and the chi-squared test for qualitative variables. Stepwise logistic regression analyses were performed to test the independent risk factors for DKA occurrence at onset. One-way analysis of variance (ANOVA) was used to compare continuous variables as factors contributing to DKA severity. Poisson regression was used to estimate the temporal trend. A probability ≤ 0.05 was used to assess statistical significance. Data were analyzed using SPSS version 28.0 software (SPSS, Chicago, IL, USA).

## 3. Results

The characteristics of patients according to the presentation at the onset of T1D are shown in Table 1.

The total number of children aged 0–14 years diagnosed with T1D in the Trentino–Alto Adige region during the period 2014–2023 was 344 (34.4/year). The incidence rate per 100,000/year ranged from 16.9 to 31.1; the peak was in 2021 during the SARS-CoV-2 pandemic (Figure 1). The average rate of DKA was 36.9%, with the lowest percentages in 2017 and 2022 (Figure 1); the frequency of DKA was lower in 2022 (vs. 2021), after the peak in new cases of T1D in 2021 (*p* = 0.040), but considering the whole 10-year period, no significant differences were found (*p* = 0.333).

Among the 344 subjects, we collected all the clinical, demographic, and socioeconomic data for a total of 272 subjects whose informed consent was obtained, and a summary is reported in Table 2.

At T1D onset, in the mean, 43% of the subjects were female (*p* = 0.015), with the lowest value in 2021 (20%) and the highest (60%) in 2018; the mean age at onset was 8.8 ± 3.9 years, without significant differences during the ten years (*p* = 0.249), and most of the individuals were 10–14 years old (45.6%, *p* < 0.0001). The onset of T1D was similar in spring, autumn, and winter and lower in summer (*p* = 0.012), without a difference in the temporal trend (*p* = 0.321). The mean HbA1c level at T1D onset was 11.8 ± 2.5% (105.7 ± 27.7 mmol/mol), without differences during the decade. The fathers’ ages increased significantly during the ten years (*p* = 0.002); the mean age of the fathers was 43.8 ± 6.8 years.

The proportion of patients with FDR with T1D was 10%; as a mean, 76.5% of the subjects with T1D onset were Italians, 12% were from the north of Africa, 8% were from the east of Europe, and 3.5 were from other countries. In 2017 and 2020, there was a higher incidence of T1D in people from North Africa (24 and 32%, respectively), and in those from East Europe (21%) in 2023. The residence at disease onset among the two provinces was 43% in the province of Bolzano, 57% in the province of Trento, and 62% were residents in a rural area. The study cohort was divided into low, moderate, and high SES.

In Table 3, in the subcohort of 272 subjects with T1D, we compared those who were displaying the onset of DKA (No. 106) to those without DKA (No. 166).

Cerebral edema and death following DKA treatment was not observed. The patients presenting with DKA were younger than the patients without DKA at diagnosis (8.0 ± 4.0 years vs. 9.4 ± 3.8) (*p* = 0.004). The DKA presence was higher in the group aged 0–4 years (*p* = 0.034), and there was no seasonality in the DKA onset. The mean HbA1c level for the patients presenting with DKA was 113.4 ± 23.4 mmol/mol (12.7 ± 2.0%) and higher than that for the patients not affected by DKA (101.7 ± 29.3 mmol/mol, 11.5 ± 2.6%, *p* < 0.0001). People with an FDR with T1D did not have higher rates of DKA (*p* = 0.527), as did people with East European or North African ethnicities (*p* = 0.411). The parents’ age and SES were similar in the two groups.

The logistic regression analysis showed that the independent characteristics of the patients with DKA were the younger age and higher values of HbA1c at T1D onset.

## 4. Discussion

In Italy, T1D incidence is very heterogeneous, and a national epidemiological registry is unavailable due to the difficulty of obtaining reliable data from all regions [21]. In the northern Italian region of Trentino–Alto Adige, the incidence of T1D, the frequency and severity of DKA at diabetes onset, and the temporal trend in the last decade have not yet been studied. This study aimed to collect these data and investigate possible factors associated with DKA to target resource allocation to reduce the rate and severity of this complication.

We found that the incidence of T1D in the Trentino–Alto Adige region in the population aged 0–14 years was 21.5/100,000 person/years, and this rate is similar in other areas in the north of Italy, such as Veneto (19.7/100,000) [5]. We registered a marked increase (+37%) during the COVID-19 pandemic (31.1/100,000 during 2021), and a recent metanalysis confirmed a +14% increase in T1D onset in 2020 during the pandemic period and +35% in 2021 [22]. The rise in incidence might be due to virus-related illness precipitating clinical diagnosis of T1D rather than a change in the risk of developing T1D, which often takes years [21]. However, during this decade, we did not register a significant increase in the T1D incidence in youths, unlike the average increase of 3–4% per year seen in the incidence reported in other populations [23] but not in others such as Finland and Norway, which showed a deceleration in the incident rate of T1D [24].

Most subjects at T1D onset were 10–14 years old, as reported in many populations [23], and we did not find a decreasing peak age of incidence during the ten years of the study observation period, in contrast to other cohorts [25]. During this decade, we found differences in the female/male proportions, with a mean higher percentage of males with T1DM, as reported in other populations [26], without an evident explanation, even though it is usually expected that autoimmune disease is more likely to affect women [27]. Data from the SEARCH study in the United States recently showed the highest increase in the rate of T1D in Black and Hispanic youth, compared to that in non-Hispanic White youth; in contrast, in our cohort, the incidence in the immigrant background did not increase during the decade [28].

The socioeconomic status and rural or urban residence at T1D onset were not different among the subjects at T1D onset, meaning that these parameters related to the role of environmental determinants had no important impact on the pathogenesis of T1D in our cohort. Instead, we found that the incidence of T1D onset was similar in spring, autumn, and winter and lower in summer, as in other populations, which was probably associated with higher rates of viral infections as a trigger [29].

The second aim of this study was to collect data on DKA incidence and its determinants. We found a high mean incidence (36.9%), which is still high in 2023, similar to the data reported in other studies from Italy [13]. DKA presented a nadir in 2017 and 2022 in our cohort, probably due to the increased awareness of T1D in youths due to the higher incidence in the years just before. Still, it has remained the same in our country over the last decade.

The younger patients (0–4 years) were more likely to present with DKA in our region; these data have been previously reported. This is due to more aggressive β-cell destruction and less well-developed compensatory mechanisms, which could result in the faster development of acidosis [30].

As previously reported, the children diagnosed with DKA had a higher HbA1c level at T1D onset [9,31], indicating a longer duration of the preclinical disease state. Instead, having an FDR with T1D did not protect against DKA, and a lower SES and level of parental education and ethnicity were not associated with a higher risk of DKA. We expected that these factors, related to a lack of awareness, language and cultural barriers, and difficulties in accessing healthcare, could impact higher rates of DKA [15]. The COVID-19 pandemic did not increase DKA rates in our study. In contrast, in other studies in 2020, DKA rates increased as the diagnostic process of T1D was delayed due to the closure of non-COVID-19 services and parental fears over contracting SARS-CoV-2 infection delayed access to healthcare services [6].

This study presents some strengths: (i) for the first time, we report data on T1D and DKA incidence in our region over ten years; (ii) electronic health records, matched with data from the Hospital Discharge Registry and User Exempt Registry, allow us to overcome the problem of missing incident cases; (iii) environmental determinants of DKA are explored well due to the availability of demographic data, ethnicity, parents’ age, and socioeconomic status.

However, this study presents a few limitations: (i) not all the subjects gave consent to collect the sociodemographic and clinical data; (ii) SARS-CoV-2 molecular test data are not available, as during the pandemic, we used antigenic tests, protocols, and assays were heterogeneous between centers and changed rapidly, and as a consequence, we avoided an analysis of any causal association between the virus and the increase in T1D incidence, but this was not an aim of this study.

For clinical practice, the systematic collection of T1D cases, reporting some basic information, can be an instrument to monitor the spatial and temporal trends in diabetes and DKA frequency. This surveillance allows for the implementation of prevention strategies at the primary level of care, such as rapidly diagnosing T1D with simple blood glucose tests and preventing DKA. Instead, additional data on environmental factors could help etiological research and studies on the determinants of T1D [27].

## 5. Conclusions

In conclusion, for clinical practice, this study reported a stable incidence rate of T1D onset over the decade in our region, affecting more children aged 10–14 years old, without a correlation with environmental factors, except for viral infections. DKA rates still remain high, and younger children are more affected. Prevention information campaigns targeted at younger children could increase awareness and effectively promote the prompt identification of T1D symptoms, reducing the rate and severity of DKA.

## Figures and Tables

**Figure 1 jpm-14-01055-f001:**
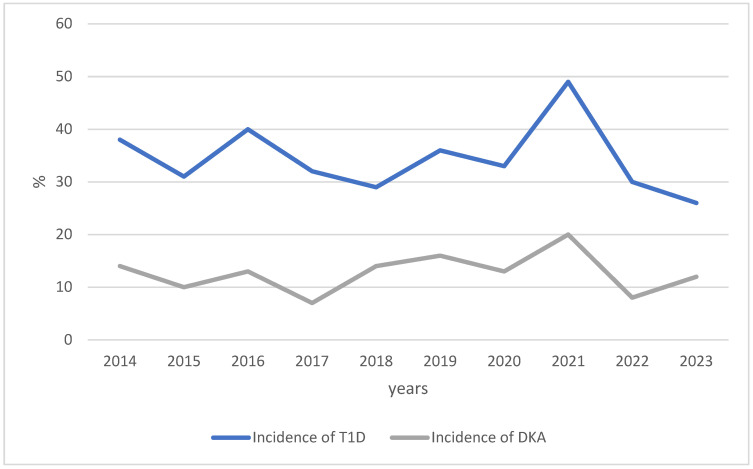
Number of incident cases of T1D and DKA in Trentino–Alto Adige (0–14 yrs, 2014–2023). Abbreviations: DKA: diabetic ketoacidosis, T1D: type 1 diabetes.

**Table 1 jpm-14-01055-t001:** Characteristics of patients according to the presentation at the onset of T1D (No. 344).

Year	2014	2015	2016	2017	2018	2019	2020	2021	2022	2023	Total
Number of new T1D onset	38	31	40	32	29	36	33	49	30	26	344 (34.4/year)
Province of Bolzano	22	14	26	14	19	16	15	34	17	20	197 (57%)
Province of Trento	16	17	14	18	10	20	18	15	13	6	147 (43%)
Population 0–14 years in Trentino Alto Adige *	164,707	163,866	162,943	162,139	161,517	160,642	159,043	157,508	156,227	154,290	
Incidence in Trentino Alto Adige No./100,000/year	23.1	18.9	24.6	19.7	18.0	22.4	20.8	31.1	19.2	16.9	21.46 (95% C.I. 19.19–23.73)
With DKA	14	10	13	7	14	16	13	20	8	12	127
Province of Bolzano	8	7	11	1	10	5	4	15	5	8	74 (58%)
Province of Trento	6	3	2	6	4	11	9	5	3	4	53 (42%)
(%)	36.8	32.2	32.5	21.8	48.3	44.4	39.4	40.8	26.7	46.1	36.9

* Mean population for each year: (population on 1st January of the year + population on 1st January of following year)/2. Source: demo.istat.it. Abbreviations: CI: confidence interval, DKA: diabetic ketoacidosis, T1D: type 1 diabetes.

**Table 2 jpm-14-01055-t002:** Clinical, demographic, and socioeconomic data for the subjects who gave consent (No. 272). Data are presented as frequencies (and %) and mean ± DS, and statistical analysis was performed to evaluate the 10-year trend.

Year	2014	2015	2016	2017	2018	2019	2020	2021	2022	2023	Total	*p*-Value
No. of subjects analyzed/ total (% of the total No.)	25/38 (66)	24/31 (77)	25/40 (62)	25/32 (78)	25/29 (86)	31/36 (86)	28/33 (85)	40/49 (82)	25/30 (83)	24/26 (92)	272/344 (79)	
Female (%)	7 (28)	9 (37)	12 (48)	14 (56)	15 (60)	14 (45)	15 (53)	8 (20)	11 (44)	11 (46)	116 (43)	**0.040**
Mean age at onset (year)	8.8 ± 4.1	7.9 ± 4.3	9.3 ± 3.5	8.1 ± 4.0	7.2 ± 4.4	8.7 ± 3.6	9.8 ± 3.6	8.9 ± 4.3	9.3 ± 3.2	10.4 ± 3.6	8.8 ± 3.9	0.249
Age group (n)												**0.023**
0–4 years old	4	8	2	7	11	6	2	10	2	3	55 (20.2%)	
5–9 years old	10	7	12	8	7	12	9	11	14	4	93 (34.2%)	
10–14 years old	11	9	11	10	7	13	17	19	10	17	124 (45.6%)	
Season of T1D diagnosis												0.321
Spring (March, April, May)	5	5	5	10	4	9	9	9	10	5	71 (28%)	
Summer (June, July, August)	6	3	3	2	7	8	2	7	4	8	50 (18%)	
Autumn (September, October, November)	4	7	7	3	5	9	9	14	7	4	69 (25%)	
Winter (December, January, February)	10	9	10	10	9	5	8	10	4	7	82 (29%)	
HbA1c (mmol/mol)	108.1 ± 26.2	93.2 ± 31.1	100.9 ± 25.2	108.7 ± 30.7	98.5 ± 26.4	114.9 ± 29.2	112.5 ± 24.1	104.3 ± 26.8	104.8 ± 19.6	112.0 ± 31.3	105.7 ± 27.7	0.149 °
HbA1c (%)	11.4 ± 2.0	10.7 ± 2.8	11.4 ± 2.3	11.8 ± 2.5	11.3 ± 2.5	13.0 ± 2.9	12.3 ± 2.3	12.6 ± 2.5	11.7 ± 1.8	12.9 ± 1.3	11.8 ± 2.5	0.128 °
Father’s age	46.0 ± 4.4	39.3 ± 7.3	43.1 ± 7.7	45.52 ± 7.6	42.3 ± 6.6	43.4 ± 4.5	41.3 ± 6.0	43.4 ± 6.9	45.8 ± 4.9	48.2 ± 8.9	43.8 ± 6.8	**0.002 ** ^†^
Mother’s age	43.1 ± 4.6	37.6 ± 6.6	40.3 ± 6.1	39.96 ± 5.9	38.2 ± 5.6	40.0 ± 5.1	39.7 ± 5.05	41.2 ± 7.1	41.5 ± 5.7	42.8 ± 6.7	40.3 ± 5.9	0.234 ^†^
FDR with T1D (%)	1 (4)	2 (8)	1 (4)	3 (12)	4 (16)	7 (22)	3 (11)	3 (7)	1 (4)	3 (12)	28 (10)	0.362 *
Father	0	0	0	1	3	2	1	1	0	0		
Mother	0	0	0	0	0	2	1	1	0	1		
Brother	0	1	1	2	1	2	0	0	0	0		
Sister	1	1	0	0	0	1	1	1	1	2		
Ethnicity (%)												0.077 **
Italians	21 (84)	17 (71)	23 (92)	16 (64)	23 (92)	24 (77)	18 (64)	33 (83)	16 (64)	18 (75)	209 (76.8)	
East Europe	1 (4)	3 (13)	1 (4)	2 (8)	0 (0)	4 (13)	1 (4)	2 (5)	3 (12)	5 (21)	22 (8.2)	
North Africa	2 (8)	1 (4)	0 (0)	6 (24)	1 (4)	3 (10)	9 (32)	5 (12)	3 (12)	1 (4)	32 (11.3)	
Other	1 (4)	3 (12)	1 (4)	1 (4)	1 (4)	0 (0)	0 (0)	0 (0)	3 (12)	0 (0)	10 (3.6)	
Residence at the diagnosis (%)												0.064
Urban	12 (48)	10 (42)	9 (36)	13 (52)	6 (24)	6 (19)	9 (32)	13 (32)	12 (48)	14 (58)	104 (38)	
Rural	13 (52)	14 (58)	16 (64)	12 (48)	19 (76)	25 (61)	19 (68)	27 (68)	13 (52)	10 (42)	168 (62)	
SES												0.151
High	6	5	7	7	8	10	4	13	8	10	78 (29%)	
Moderate	9	11	12	7	8	14	5	15	5	8	94 (35%)	
Low	10	8	6	10	8	7	18	11	12	6	96 (36%)	

* *p*-value calculated between patients with and without FDR with T1D; ** *p*-value calculated between participants of Italian and foreign ethnicity; ° *p*-value based on the Kruskal–Wallis test; ^†^
*p*-value based on the one-way ANOVA; Abbreviations: FDR: first-degree relative, T1D: type 1 diabetes.

**Table 3 jpm-14-01055-t003:** Comparison between the group of subjects with DKA and without DKA (No. 272).

Characteristics	With DKA (No. 106)	Without DKA (No. 166)	*p*-Value
Female (%)	51 (48)	65 (39)	0.18 *
Age	8.0 ± 4.0	9.4 ± 3.8	**0.004** °
Age groups (%) 0–4 years old	29 (27.4)	26 (15.7)	**0.034 ***
5–9 years old	37 (34.9)	56 (33.7)	
10–14 years old	40 (37.7)	84 (50.6)	
HbA1c % mmol/mol	12.7 ± 2.0 113.4 ± 23.4	11.5 ± 2.6 101.7 ± 29.3	**<0.0001** ° **<0.0001** °
pH	7.1 ± 0.12	7.4 ± 0.06	**<0.0001** °
Season (%) Spring Summer Autumn Winter	25 (23.5) 18 (17.0) 28 (26.5) 35 (33.0)	46 (27.7) 32 (19.3) 41 (24.7) 47 (28.3)	0.763 *
FDR with T1D (%)	9 (8.5)	18 (10.8)	0.527 *
Ethnicity (%) Italians East Europe North Africa Other	80 (75.5) 10 (9.5) 10 (9.5) 6 (5.5)	129 (77.7) 12 (7.2) 21 (12.7) 4 (2.4)	0.411 *
Father’s age (year)	42.7 ± 0.1	44.5 ± 6.6	0.024 °
Father’s age class (%) <40 y 40–49 y ≥50 y	No. 99 32 (32.3) 49 (49.5) 18 (18.2)	No. 162 35 (21.6) 93 (57.4) 34 (21.0)	0.157 *
Mother’s age (year)	39.7 ± 6.45	40.6 ± 5.5	0.245 °
Mother’s age class (%) <40 y 40–49 y ≥50 y	No. 106 50 (47.2) 52 (49.0) 4 (3.8)	No. 164 71 (43.3) 88 (53.7) 5 (3.0)	0.748 *
Family SES (%) Low Moderate High	No. 103 42 (40.8) 36 (34.9) 25 (24.3)	No. 165 54 (32.7) 58 (35.2) 53 (32.1)	0.288 *

* *p*-value based on chi-squared test; ° *p*-value based on the Kruskal–Wallis test; Abbreviations: FDR: first-degree relative, SES: socioeconomic status, T1D: type 1 diabetes.

## Data Availability

Data are available on request.

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
