# Peer review of "Incidence of Type 1 Diabetes in Children Aged 0–14 Years in Trentino–Alto Adige Region and Determinants of Onset with Ketoacidosis"

_jpm, 2024, doi:10.3390/jpm14101055_

Round 1
Reviewer 1 Report
Comments and Suggestions for Authors
Thank you for the opportunity to review the manuscript of Fanti et al aiming to analyze the variation in the incidence of type 1 diabetes and the onset with ketoacidosis in in children aged 0-14 years in 2 Trentino Alto-Adige Region. I congratulate the authors for the interesting subject and the quality of the research and of the manuscript.
The Introduction is well written setting the ground for the research. The methodology is appropriate and described in detail, including the statistical analysis, and the results are clearly presented and discussed in the context of existing literature.
I have some minor comments that would help the reader understand the data:
1. Please use different colors to show incidence of DKA and T1D in Figure 1. Now is difficult to discriminate between them.
2. Also, define abbreviations used in tables and figure in their legends. They should stand alone and be easily understood without reading the article text.
Author Response
Point by point replay
Reviewer #1:
Thank you for the opportunity to review the manuscript of Fanti et al aiming to analyze the variation in the incidence of type 1 diabetes and the onset with ketoacidosis in in children aged 0-14 years in Trentino Alto-Adige Region. I congratulate the authors for the interesting subject and the quality of the research and of the manuscript.
The Introduction is well written setting the ground for the research. The methodology is appropriate and described in detail, including the statistical analysis, and the results are clearly presented and discussed in the context of existing literature.
I have some minor comments that would help the reader understand the data:
We are very thankful for the positive comments of the reviewer. We greatly appreciate the opportunity to submit a revised version of our manuscript.
Please use different colors to show incidence of DKA and T1D in Figure 1. Now is difficult to discriminate between them.
Answer: thank you very much for your suggestion. We provide Figure 1 in colour.
- Also, define abbreviations used in tables and figure in their legends. They should stand alone and be easily understood without reading the article text.
Answer: we agree with your suggestion. We added abbreviations into Figure legends.
Reviewer 2 Report
Comments and Suggestions for Authors
The study investigates the incidence and trends of Type 1 Diabetes (T1D) and diabetic ketoacidosis (DKA) among youths aged 0-14 years in the Trentino Alto Adige region of Italy from 2014 to 2023. The study found an incidence of T1D in the Trentino Alto Adige region comparable to other northern Italian areas. The mean age of the participants was 8.8 years. We note a high incidence of DKA.
I propose some English editing to increase the quality of the paper.
I have some minor comments that should be addressed further:
Abbreviations are usually defined at the first use in the abstract and in the main text.
„ children and adolescents aged between 1-14.99 years” 14.99 years here are unnecessary. Consider rounding the number. The same comment applies to the whole manuscript, where age should be a round number of years.
The following fragment is a history: „In Italy, starting in 1997, a Registry of Childhood Type 1 Diabetes Mellitus in Italy (RIDI) was established to coordinate regional registries, but only a few regions (Liguria, Marche, Umbria, Lazio, Abruzzo, Campania, and Sardinia) and provincial registries (Trento, Turin, Pavia, Modena, and Florence-Prato) were included. In the Trentino-Alto-Adige Region, no institutional Regional registers are available, and only two centers belonging to the Italian Society of Pediatric Endocrinology and Diabetology (ISPED) network [17] usually receive all the new cases of youths with pediatric diabetes: the Center of Bolzano and the one of Trento, and register the patients in electronic health records.”. Too many details, consider rephrasing the ideas.
The International Society for Pediatric and Adolescent Diabetes (ISPAD) guidelines are not cited.
Please add labels on the pictures describing ox and oy.
The discussion section should start with a research summary stating the specific findings and how these fill a gap in the literature.
Comments on the Quality of English Language
There are many inconsistencies in English grammar.
Author Response
Reviewer #2:
The study investigates the incidence and trends of Type 1 Diabetes (T1D) and diabetic ketoacidosis (DKA) among youths aged 0-14 years in the Trentino Alto Adige region of Italy from 2014 to 2023. The study found an incidence of T1D in the Trentino Alto Adige region comparable to other northern Italian areas. The mean age of the participants was 8.8 years. We note a high incidence of DKA.
We are very thankful for the valuable comments of the reviewer. We greatly appreciate the opportunity to submit a revised version of our manuscript. The criticisms prompted us to improve our work.
In the first paragraph of the discussion, we compared the incidence of T1D and DKA with other Italian regions and other Countries.
I propose some English editing to increase the quality of the paper.
Answer: thank you for your suggestion. English has been edited.
I have some minor comments that should be addressed further:
Abbreviations are usually defined at the first use in the abstract and in the main text.
Answer: we agree with your comment. Abbreviations have been checked.
„ children and adolescents aged between 1-14.99 years” 14.99 years here are unnecessary. Consider rounding the number. The same comment applies to the whole manuscript, where age should be a round number of years.
Answer: we agree with your comment.
The following fragment is a history: „In Italy, starting in 1997, a Registry of Childhood Type 1 Diabetes Mellitus in Italy (RIDI) was established to coordinate regional registries, but only a few regions (Liguria, Marche, Umbria, Lazio, Abruzzo, Campania, and Sardinia) and provincial registries (Trento, Turin, Pavia, Modena, and Florence-Prato) were included. In the Trentino-Alto-Adige Region, no institutional Regional registers are available, and only two centers belonging to the Italian Society of Pediatric Endocrinology and Diabetology (ISPED) network [17] usually receive all the new cases of youths with pediatric diabetes: the Center of Bolzano and the one of Trento, and register the patients in electronic health records.”. Too many details, consider rephrasing the ideas.
Answer: thank you for your comment. The statement has been rephrased.
The International Society for Pediatric and Adolescent Diabetes (ISPAD) guidelines are not cited.
Answer: Please see ref. n. 18.
Please add labels on the pictures describing ox and oy.
Answer: thank you for your suggestion. Labels have been added to the figure 1.
The discussion section should start with a research summary stating the specific findings and how these fill a gap in the literature.
Answer: thank you for your suggestion. Thanks, we started the discussion section as you suggested.
Comments on the Quality of English Language
There are many inconsistencies in English grammar.
Answer: thank you for your suggestion. English has been edited.